# Modulation of Mesenchymal Stem Cells for Enhanced Therapeutic Utility in Ischemic Vascular Diseases

**DOI:** 10.3390/ijms23010249

**Published:** 2021-12-27

**Authors:** Sally L. Elshaer, Salma H. Bahram, Pranav Rajashekar, Rajashekhar Gangaraju, Azza B. El-Remessy

**Affiliations:** 1Department of Pharmacology and Toxicology, Faculty of Pharmacy, Mansoura University, Mansoura 35516, Egypt; dr_s_elshaer@mans.edu.eg; 2Department of Clinical Pharmacy and Pharmacy Practice, Faculty of Pharmacy, Misr International University, Cairo 19648, Egypt; salma.bahram@miuegypt.edu.eg; 3Department of Ophthalmology, University of Tennessee Health Science Center, Memphis, TN 38163, USA; prajashe@uthsc.edu (P.R.); 4Departments of Ophthalmology & Anatomy and Neurobiology, University of Tennessee Health Science Center, Memphis, TN 38152, USA; sgangara@uthsc.edu (R.G.); 5Department of Pharmacy, Doctors Hospital of Augusta, Augusta, GA 30909, USA

**Keywords:** mesenchymal stem cells (MSCs), inflammation, angiogenesis, repair, ischemic heart disease, ischemic retina, peripheral limb ischemia, wound healing, scar, immune modulation

## Abstract

Mesenchymal stem cells are multipotent stem cells isolated from various tissue sources, including but not limited to bone marrow, adipose, umbilical cord, and Wharton Jelly. Although cell-mediated mechanisms have been reported, the therapeutic effect of MSCs is now recognized to be primarily mediated via paracrine effects through the secretion of bioactive molecules, known as the “secretome”. The regenerative benefit of the secretome has been attributed to trophic factors and cytokines that play neuroprotective, anti-angiogenic/pro-angiogenic, anti-inflammatory, and immune-modulatory roles. The advancement of autologous MSCs therapy can be hindered when introduced back into a hostile/disease environment. Barriers include impaired endogenous MSCs function, limited post-transplantation cell viability, and altered immune-modulatory efficiency. Although secretome-based therapeutics have gained popularity, many translational hurdles, including the heterogeneity of MSCs, limited proliferation potential, and the complex nature of the secretome, have impeded the progress. This review will discuss the experimental and clinical impact of restoring the functional capabilities of MSCs prior to transplantation and the progress in secretome therapies involving extracellular vesicles. Modulation and utilization of MSCs–secretome are most likely to serve as an effective strategy for promoting their ultimate success as therapeutic modulators.

## 1. Introduction

### 1.1. Potential Therapeutic Use of Mesenchymal Stem Cells (MSCs)

Mesenchymal stem cells (MSCs) are non-hematopoietic, multipotent stem cells derived from the middle germ layer, mesoderm. They can also be referred to as “mesenchymal stromal cells” because of their origin [1]. MSCs can migrate to the pathological site due to disease-related secretion of biologically active regenerative and immunomodulatory factors [2]. MSCs are typically sourced from a wide array of cell categories, including but not restricted to the umbilical cord, adipose, and bone marrow tissue. Bone marrow (BM) is one of the most widely used sources of MSCs, in which they constitute 0.001% to 0.01% of mononuclear cells [3]. MSCs can also be isolated from other sources, including umbilical cord blood, placenta, adipose tissue, and skin. One significant advantage of their isolation is that the process is not invasive, nor does it raise ethical or legal objections [4]. An essential feature of MSCs is their low immunogenicity, allowing allogeneic transplantation in the clinical setting [5,6]. Of note, new research has shown that MSCs can elicit an immune response, activating T lymphocytes under the influence of interferon-gamma (IFN-γ) [7,8]. 

The mobilization of MSCs is usually triggered by a plethora of secreted factors from the damaged cells. When MSCs are activated, they can migrate and settle in the appropriate pathological site to exert reparative action (reviewed in [9,10]). The secreted factors can be classified into growth factors, cytokines, and chemokines. Examples of growth factors are vascular endothelial growth factor (VEGF), stromal-derived factor 1 (SDF-1), granulocyte-colony stimulating factor (G-CSF), erythropoietin, angiopoietin 2, placental growth factor (PlGF), platelet-derived growth factor (PDGF), Stem cell factor (SCF), insulin-like growth factor-1 (IGF-1), Epidermal growth factor (EGF), and hepatocyte growth factor (HGF). The cytokines include interleukins (IL) such as IL-1b, IL-2, IL-3, IL-6, IL-8, and tumor necrosis factor-α (TNF-α) and chemokines include CCL5 and CCL22 [9,10]. As detailed in the following sections, the protective mechanism of MSCs could be attributed partially to their direct differentiation and replacement of damaged tissues; human MSCs differentiated into hepatocyte-like cells in vitro and exerted hepatocyte functions in vivo in mice [11]; umbilical cord MSCs differentiated into beta-like pancreatic islet cells in vitro [12] and rat MSCs could differentiate into neuron-like cells in vitro [13]. Nevertheless, the protective mechanisms of MSCs are primarily mediated through paracrine properties, as discussed in the following section.

### 1.2. MSCs Secretome

MSC’s secretome is defined as a complex mixture of soluble fractions (represented by growth factors and cytokines) and a vesicular fraction constituting microvesicles and exosomes (depicted in Figure 1). Exosomes are essential for transferring proteins and genetic material to other cells (reviewed in [14]). A growing body of evidence supports the role of MSCs’ secretome facilitating cell proliferation, survival, and differentiation and physiological/functional outcomes using in vitro and in vivo models of disease. There is a long list of secreted growth factors, including transforming growth factor-beta (TGF-β), VEGF, IGF-1, SDF-1, fibroblast growth factor (FGF), nerve growth factor-beta (NGF-β), G-CSF, HGF, and EGF (reviewed in [15]). In addition, MSCs secrete cytokines and chemokines, including CXCL12 (SDF-1), CCL2, CCL5 [16]. MSCs secretome obtained for therapeutic application can be tailored or modified to the desired cell-specific effects. The secretome of MSCs from different tissue sources or the passage number of MSCs can exhibit distinct secretory profiles and exosomal compositions [15]. For example, MSCs isolated from adipose tissue have a more substantial impact on axonal growth when compared to MSCs isolated from bone marrow. Nevertheless, in a different study by the same group, cellular passaging did not affect the secretome composition or function assessed by its ability to support post-natal neuronal survival and promote axonal growth [17,18].

More recently, it was discovered that MSCs could modify the microenvironment through the release of extracellular vesicles (EV). Extracellular vesicles secreted by MSCs can be mainly classified into exosomes, micro-vesicles, and apoptotic bodies [19], of which exosomes are currently the most thoroughly described subtype. Exosomes, spherical in shape and with a two-layer lipid film, 30 to 120 nm in size, are formed by the convexity of endosomal membranes in late endosomes, resulting in the formation of multi-alveolar bodies. By connecting with the cytoplasmic membrane, cells release their content to the outside in the form of exosomes [20]. Exosomes are characterized by the presence of many proteins, including annexin, tetraspanin (CD63, CD81, CD9), heat shock proteins, Clarins, caveolin, exosome specific proteins such as Alix and tumor susceptibility gene 101 (TSG 101), and proteins typical of cells from which they come [21]. In addition, exosomes contain characteristic lipids, including fragments of lipid raft; GM1 ganglioside, cholesterol, ceramides, sphingomyelin, and phosphatidylserine, as well as nucleic acids; mRNA and miRNA [22]. Synthesis and secretion of MSCs-exosomes are complex and vary depending on external stimuli such as hypoxia or inflammation. Wnt and mTOR signaling pathways have been reported to play a role in exosome secretion [23]. In addition, MSCs and their secreted exosomes were shown to express common surface antigens, including CD9, CD29, CD44, and CD89 [24]. Growing evidence supports the involvement of MSCs-derived exosomes in antigen presentation and immunological response, angiogenesis, coagulation, and programmed cell death [25]. 

### 1.3. The Potential Use for MSCs and Cell-Free Therapy in Ischemic Diseases

The advancement of autologous MSCs therapy or its secretome can be hindered when introduced back into a hostile or a disease environment. A few examples are impaired endogenous MSCs function, limited post-transplantation cell viability, and altered MSC immunomodulatory efficiency. Thus, restoring the functional capabilities of these impaired MSCs prior to transplantation or utilizing the secretome is an effective strategy for promoting their ultimate success as repair mediators. Exosomes, a cell-free therapeutic, have been the most frequently used form of injected MSCs-derived vesicles [4]. However, their poor cell engraftment and survival have mainly limited the protective potential of MSCs for various ischemic diseases. As such, attempts have been made to enhance the cellular and therapeutic effects of MSCs. In the current review, we will discuss preclinical and clinical studies involving the use of modulated MSCs and their paracrine exosomal secretion in various ischemic models, in particular, the use of MSCs in ischemic heart and retinal diseases and ischemic limbs and chronic wound healing. 

## 2. Ischemic Heart Diseases

### 2.1. Clinical Scope and Need for New Therapeutics for Ischemic Heart Diseases

Ischemic heart disease (IHD) is caused by narrowed coronary arteries that supply blood to the heart muscle. The narrowing can be caused by a blood clot, constriction of blood vessels, or, most commonly, atherosclerosis. If blood flow to the heart muscle is completely blocked, the heart muscle cells die, resulting in a heart attack or myocardial infarction. Patients with IHD usually experience discomfort during exercise or emotional stress, in a disease known as Angina Pectoris [26]. Globally, ischemic heart diseases (IHD) affect 1.72% of the world’s population and 126 million individuals. Nine million deaths were caused by IHD globally as of 2020 [27]. There have been tremendous advances in medicine and surgery to lower cardiovascular disease mortality; however, they merely serve as transient “delayers” of an inevitably progressive disease process that carries significant morbidity. Current treatment guidelines recommend antianginal therapy to control symptoms before considering coronary artery revascularization. First choice drugs are beta-blockers, calcium channel blockers, and short-acting nitrates [26]. However, patients encounter serious adverse effects while on pharmacologic treatment for IHD, i.e., bradycardia, hypotension, bronchospasm, fatigue, and blunting of the tachycardia response to hypoglycemia, resulting in postural hypotension and decreased cardiac contractility [26]. In addition, the initial ischemic insult is not entirely resolved. Due to the heart’s limited self-regenerative capacity, identification of optimal cell-based therapy is essential to assist myocardial self-repair and restore cardiac function [28]. Several cell-based strategies have been explored for cardiac regeneration, including modulated MSCs or their paracrine secretions. The following sections will highlight critical studies that implemented MSCs for IHD. In addition, completed and ongoing clinical trials will also be discussed. 

### 2.2. Preclinical Studies of Modulated MSCs in IHD Animal Models

The effectiveness of transplanted bone marrow-MSCs for cardiac repair has been limited; thus, several strategies for optimizing MSCs’ therapeutic potential have been investigated. Such strategies include ischemic preconditioning, genetic modification, and optimized MSCs treatment before injection. In 2016, hypoxia preconditioning was reported to improve the effectiveness of MSCs transplantation for the treatment of myocardial infarction in Cynomolgus monkeys without increasing the occurrence of arrhythmogenic complications [29]. Overexpression of CCR1 increased MSCs viability, migration, engraftment, and capillary density in the injured myocardium [30]. Infusion of MSCs derived from TNF receptor-1 (TNFR-1) knockout mice restores cardiac function after ischemic-reperfusion injury, suggesting that TNFR-1 negatively modulates MSCs’ paracrine action [31]. SDF-1 interaction with its receptor CXCR-4 is vital for preserving the myocardium. SDF-1-overexpressing MSCs improved myocardial function primarily through preservation, not the regeneration of cardiac myocytes within the infarct zone [32]. Similarly, Bcl-2 engineered MSCs inhibited apoptosis and improved heart function in a rat model of acute myocardial infarction [33]. Yan W et al., 2020, reported that MSCs over-expressing N-Cadherin is protective against ischemic heart injury through β-Catenin-Dependent Manner [34]. Regarding optimizing MSCs with prior treatment protocols, Guo J et al., 2008, reported that pretreatment of MSCs with IGF-1 enhanced engraftment in the infarct-border zones, increased capillary density, and attenuated cardiac dysfunction, left ventricular chamber enlargement, and scar formation in a rat model of myocardial infarction [35]. Additionally, priming MSCs with a cocktail of growth factors including FGF-2, IGF-1, and bone morphogenetic protein-2 (BMP-2) improved cardiac function in a rat model of myocardial infarction [36]. Trimetazidine is an anti-ischemic drug used for treating angina. Wisel S et al., 2009, reported that hearts transplanted with trimetazidine-preconditioned MSCs improved cardiac function post-LAD ligation with overexpression of Akt and Bcl-2 [37]. MSCs are known for their paracrine actions that contribute to cardiac functional recovery by aiding endogenous repair mechanisms [38]. The mechanism could be attributed to activation of the JAK-signal transducer and STAT3 signaling pathway and the subsequent increase in production of host-tissue HGF and VEGF that mediate activation of endogenous cardiac repair mechanisms [39]. Soluble factors released by MSCs not only stimulate cardiomyocyte regeneration and angiogenesis but further support antifibrosis and antiapoptotic activity as well as inhibit a pro-inflammatory state [39]. MSCs engineered to overexpress survivin protein, an X-linked inhibitor of apoptosis family members, have shown a better prognosis in a rat model of myocardial infarction (MI) [40]. The protective effects were attributed to paracrine secretion of VEGF that increased capillary density, reduced the infarct size by and improved cardiac function at 28 days after MI [40].

### 2.3. Preclinical Studies of MSCs Exosomes in IHD Animal Models

In 2010, Lai et al. were the first to report the myocardial protective potential of MSCs exosomes in a mouse model of myocardial ischemia/reperfusion injury [24]. MSCs-derived exosomes have attracted more attention for attenuating ischemic heart diseases, a mechanism that can be partially attributed to the exosome content of microRNAs (miRNAs). The miRNAs secreted by MSCs exosomes are single-stranded non-coding RNAs that can enter into the recipient cells, degrade the target mRNA, or inhibit the mRNA-protein translation process [41]. BM-MSCs-derived exosomal miR-150–5p was found to attenuate apoptosis of cardiomyocytes and improve the cardiac function of myocardial ischemia in mice via targeting Bcl-associated X protein (Bax) [42]. Furthermore, exosomes derived from MSCs were superior to MSCs in repairing injured myocardium by inhibiting fibrosis and inflammation in the rat model of myocardial infarction. The mechanism was attributed to higher miR-29 and miR-24 and lower miR-21, miR-34, and miR-378 [43]. Luther KM et al., 2018, reported the cardioprotective potential of MSCs exosomes through their content of miR-21a-5p, which can downregulate the expression of the pro-apoptotic gene products PDCD4, PTEN, Peli1, and FasL in the myocardium [44]. The protective potential of miR-21a-5p was further supported by Mayourian J et al., 2018 [45]. Additional miRNAs in MSCs exosomes contributing to their cardio-protection include MiR-22 as reported by Feng Y et al., 2014 [46]; MiR-221/222 as reported by Lai T et al., 2020, [47] and MiR-185 as reported by Li Y et al., 2020 [48]. 

Ischemic preconditioning of MSCs can potentiate their paracrine protective effect on the myocardium [29], where hypoxic MSCs-derived exosomes were shown to possess higher angiogenic activity than normoxic exosomes [23]. Supporting this finding, Park H et al., 2018, reported the protective effect of hypoxic human MSCs via miRNA-26a in an ischemia-reperfusion injury model in rats [49]. Exosomes secreted from GATA-4 overexpressing MSCs served as a reservoir of the anti-apoptotic miR-19a and increased the survival rate of rat cardiomyocytes by decreasing apoptosis and preserving mitochondrial membrane potential [50]. Hypoxic Akt-overexpressing MSCs mediated cardio-protective potential and functional improvement through upregulating genes that encode for soluble factors, including VEGF, FGF-2, HGF, and IGF-1 [51]. In addition, Akt-MSCs induced myocardial survival and repair through the paracrine release of secreted frizzled-related protein 2, which is a known regulator for the Wnt signaling pathway [52]. Likewise, Akt-modified MSCs mediated cardiac protection and functional improvement through a paracrine mechanism rather than myocardial regeneration from donor cells [51]. Exosomes derived from SDF1-overexpressing MSCs inhibited ischemic myocardial cell apoptosis and promoted cardiac endothelial microvascular regeneration in mice with myocardial infarction [53]. Overexpression of heme oxygenase-1 (HO-1)-MSCs mediates cardiac protection and functional improvement through a paracrine mechanism in a rat model of myocardial infarction [54]. 

Cardio-protection of MSCs exosomes could be related to autophagy. Autophagy can improve cell survival by degrading damaged organelles to produce ATP and reducing the infarct area; however, it can result in the death of cardiomyocytes under long-term or severe ischemia [55]. MSCs reduced autophagic flux in infarcted mouse hearts via the exosomal transfer of miR-125b [56]. Likewise, during remote ischemic preprocessing, the application of exosomes rich in miR-144 can increase P-Akt, P-GSK3β, and P-p44/42 MAPK, decreasing phosphorylated mTOR (p-mTOR) level and inducing autophagy to improve myocardial function and reduce infarct size in mice [57]. In the context of inflammation, MSCs-derived exosomes attenuated myocardial ischemia-reperfusion injury through miR-182-regulated macrophage polarization from M1 to M2 status, thus reducing the level of inflammation [58]. 

### 2.4. Clinical Studies of MSCs in IHD

To date, multiple phase I and II clinical trials have been completed using intravenous (IV), intracoronary (IC), and intramyocardial (IMC) delivery routes of MSCs for IHD (see Table 1). Hare et al., 2009, conducted a phase-I randomized, double-blinded study and reported the safety and efficacy of IV delivered allogenic human MSCs in acute myocardial infarction (MI) patients [59]. These encouraging data led to a phase II trial that is still ongoing using a similar study design. Another phase-I study conducted by Rodrigo et al., 2013, reported the feasibility and safety of autologous IMC injection of MSCs in acute MI patients [60]. AMICI trial is an ongoing phase-II clinical trial investigating the safety and efficacy of allogeneic Mesenchymal Precursor Cells (MPCs) infused IC in patients with acute STEMI. Whereas the CADUCEUS trial was a phase-I randomized, dose-escalation study that reported the therapeutic potential of IC delivered autologous cardiosphere-derived stem cells, which include MSCs-like cardiac cells, 6 months post-infusion in MI patients [61]. A follow-up to that trial published in 2014 showed the maintenance of safety outcomes and therapeutic regeneration even until one 1-year post-treatment [62]. ACCRUE study (Meta-Analysis of Cell-based CaRdiac studies) is the first prospective collaborative multinational database for patients with acute MI treated with cell therapy. The study analyzed the safety and efficacy of IC cell administration, collecting individual patient data from 12 randomized trials. The study strongly suggests that IC cell therapy may not be the optimal therapeutic strategy to benefit the patients in terms of clinical events or changes in left ventricular function [63]. 

**Table 1 ijms-23-00249-t001:** Clinical Trials that utilized MSCs in ischemic heart diseases.

Study	Clinical Trials Identifier:	Intervention	Delivery Method	Disease	Phase	Status	Reference
A randomized, double-blind, placebo-controlled, dose-escalation study of intravenous adult human MSC (prochymal) after acute myocardial infarction	NCT00114452	Allogeneic hMSCs	IV	myocardial infarction	Phase I	Completed	[59]
Safety Study of Adult Mesenchymal Stem Cells (MSC) to Treat Acute Myocardial Infarction	NCT00114452	Provacel ex vivo cultured adult MSCs	IV	Myocardial Infarction	Phase I	Completed	[59]
Intramyocardial Injection of Autologous Bone Marrow-Derived Ex Vivo Expanded Mesenchymal Stem Cells in Acute MI Patients is Feasible and Safe up to 5 Years of Follow-up	Dutch trial registry (NTR1553)	Autologous BM-MSCs	Intramyocardial injection	Acute ST-segment elevation MI		Completed	[60]
Intracoronary cardiosphere-derived cells for heart regeneration after MI (CADUCEUS): a prospective, randomized phase I trial	NCT00893360	Autologous stem cell infusion	Intracoronary infusion	Recent Myocardial Infarction Ventricular Dysfunction	Phase I	Completed	[64]
Comparison of allogeneic vs. autologous BM-MSC delivered by transendocardial injection in patients with ischemic cardiomyopathy: the POSEIDON randomized trial	NCT01087996	Auto-hMSCs Allo-hMSCs	Transendocardial Injection	LV dysfunction due to ICM	Phase I Phase II	Completed	[65]
Prospective Randomized Study of Mesenchymal Stem Cell Therapy in Patients Undergoing Cardiac Surgery (PROMETHEUS)	NCT00587990	MSC injection	Intramyocardial injection	Left Ventricular Dysfunction	Phase I Phase II	Terminated (Difficulty in recruitment.)	[66]
Adipose-derived regenerative cells in patients with ischemic cardiomyopathy: The PRECISE Trial	NCT00426868	Direct injection of (ADRCs)	Direct injection into the Left Ventricle	coronary artery disease refractory to revascularization	Phase I	Completed	[67]
Rationale and design of the first randomized, double-blind, placebo-controlled trial of intramyocardial injection of autologous BM-MSCs in chronic ischemic Heart Failure (MSC-HF Trial)	NCT00644410	Autologous BM-MSC	Intramyocardial injection	Congestive Heart Failure	Phase I Phase II	Completed	[68]
Cardiopoietic stem cell therapy in heart failure: the C-CURE (Cardiopoietic stem Cell therapy in heart failURE)	NCT00810238	BM- cardiopoietic cells	LV endocardial injection	heart failure Class II or III of ischemic origin	Phase II Phase III	Completed	[69]
Prochymal^®^ (Human Adult Stem Cells) Intravenous Infusion Following Acute Myocardial Infarction (AMI)	NCT00877903	Prochymal^®^ (Human Adult Stem Cells)	IV infusion	Myocardial Infarction	Phase II	Completed	-
Safety Study of Allogeneic Mesenchymal Precursor Cell Infusion in MyoCardial Infarction (AMICI)	NCT01781390	Mesenchymal Precursor Cells (MPC)	Intracoronary infusion	Acute Myocardial Infarction	Phase II	Active, not recruiting	-

As for chronic ischemic cardiomyopathy, several trials have been conducted. The POSEIDON randomized trial, published in 2012, showed that trans-endocardial MSCs injection, whether allogeneic or autologous, improved functional capacity, quality of life, and ventricular remodeling in patients suffering left ventricular dysfunction associated with ischemic cardiomyopathy [65]. PROMETHEUS trial, published in 2014, showed that autologous IMC MSCs injection produces concordant improvements in regional function, tissue perfusion, and fibrotic burden when administered to patients undergoing coronary artery bypass grafting [66]. The phase II randomized, placebo-controlled, double-blinded PRECISE trial demonstrated improved myocardial perfusion, exercise capacity, and cardiac function in no-option ischemic cardiomyopathy patients receiving trans-endocardial injections of adipose-derived MSCs at 6 months post-treatment. Notably, the ventricular function was preserved up to 18 months post-treatment [67]. MyStromalCell Trial is a prospective, randomized, double-blind, placebo-controlled study that used abdominal adipose tissue-derived stem cells stimulated with VEGF-A (165). They aimed to establish the safety and efficacy of this cell therapy by using functional capacity, quality of life, and cardiac imaging parameters, in patients with chronic myocardial ischemic and refractory angina [68]. The phase III C-CURE trial implemented the paradigm of lineage guidance in cell therapy for ischemic heart failure. Endomyocardial injection of autologous cardiopoietic stem cells derived from MSCs was feasible and safe with signs of benefit in chronic heart failure [69]. Another phase III trial is dedicated to evaluating the efficacy and safety of a single trans-endocardial delivery of allogeneic MPCs for the treatment of chronic ischemic cardiomyopathy. Efficacy outcomes will be assessed until 60 months post-treatment (ClinicalTrials.gov identifier: NCT02032004). 

## 3. Ischemic Retinal Diseases

### 3.1. Clinical Scope and Need for MSC Therapeutics

Retinal ischemia is characterized by the lack of blood flow to the retinal layers of the eye, leading to hypoxia. Ischemic retinopathies, including diabetic retinopathy, retinopathy of prematurity, and retinal vascular occlusion, are increasing in prevalence, represent a major economic burden, and are significant causes of vision loss and blindness worldwide [70,71,72]. Common pathways triggered by transient hypoxia and or ischemia are reactive oxygen species (ROS) formation and eventual inflammation and apoptosis. The level of retinal ischemia can denote disease severity and progression [73]. In addition, retinal hypoxia triggers the release of VEGF and inflammatory mediators. Currently, available therapies for the late stage of ischemic retinopathy include laser photocoagulation and intravitreal injection of corticosteroids and anti-VEGF. While these therapies effectively improve vision, serious limitations include invasiveness of repeated ocular injections, risk of neuronal toxicity and atrophy, and increased thromboembolism [74]. Most importantly, these treatments cannot improve retinal ischemia to achieve reperfusion, which causes relapse to therapy in 50% of patients. Therefore, there is a great need to develop therapeutics that effectively address the root cause of the problem in retinal ischemia. MSCs, due to their pro and/or anti-angiogenic and anti-inflammatory properties, can mitigate detrimental visual effects from ischemia-associated diseases. Alternatively, the use of paracrine factors and exosomes derived from MSCs avoids potential risks of improper integration of MSCs into the retinal layers of the eye.

### 3.2. Preclinical Studies of MSCs in Ischemic Retina Animal Models

Modulation of MSCs to express or lack thereof a certain protein/gene has been explored in ischemic retinal diseases. In a model of retinal ischemic injury, intravitreal delivery of bone marrow MSC after a 24 h delay following ischemic injury showed a decrease in apoptosis in the retinal ganglion cell (RGC) layer with an improved retinal function [75]. Although limited to a short course of 7 days, authors were able to show the persistence of cells in the retina, with the majority of labeled cells found in the vitreous. Overexpression of HO-1 in MSCs showed partial restoration and a lower rate of retinal thinning of the inner plexiform layer, inner nuclear layer, and the outer nuclear layer of the retina in the ischemic retina injury model in rats [76]. Several animal studies demonstrated that transplantation of MSCs can delay the progression of retinal neurodegeneration and preserve neural retinal function [77,78]; however, the vascular protection of MSCs transplanted in the ischemic retina has not been fully elucidated. Introduction of MSCs paired with eliminating the p75^NTR^ gene decreased acellular capillaries in the ischemia-reperfusion model compared to controls [79]. The p75^NTR^ receptor, also known as CD271, is a marker of BM-MSCs with enhanced differentiation capacity for bone or cartilage repair. The conditioned medium from CD271+ MSC cultures was less angiogenic than MSC that lacked CD271 [80]. In parallel, knocking down expression or activity of p75^NTR^/CD271 in BM-MSCs resulted in significant changes in its secretome evident by increases in expression of VEGF, NGF, and SDF-1 in conditioned medium (CM). Moreover, CM from MSCs that lacked expression or activity of p75^NTR^/CD271 has exerted more angiogenic response when co-cultured with human retinal endothelial cells, assessed by enhanced endothelial cell migration and alignment to tube formation and improved expression of endothelial survival markers [79]. 

Adipose-derived MSCs (ASC) have inherent functional and phenotypic overlap with pericytes lining microvessels in adipose tissue [81]. Elimination of pericyte-specific CD140b, PDGFβ, receptor protein in ASC significantly negated retinal function and their homing ability in the neural retina [82] and impeded their angiogenic potential in vitro [83]. CD146, commonly known as melanoma cell adhesion molecule, is a pericyte marker affiliated to blood vessel integrity, higher stem cell differentiation, and higher expression of cellular repair mechanisms. ASCs positive for CD146 injected intravitreally in ischemia-reperfusion injury mice enhanced retinal function mainly via improved adhesion and migration of cells over those ASCs that lack CD146 [84]. Recent evidence showed that CD146 is required for PDGFβ receptor activation in pericytes [85], possibly suggesting a similar mechanism of these molecules might be in play in ASCs. While pericyte differentiation of MSCs is an attractive strategy, Zhang et al. showed the intravitreal injection of neural stem cells originating from human umbilical cord-derived mesenchymal stem cells decreased blood vessel leakage and increased RGC survival in diabetic rats [86]. Recently, intravitreal delivery of induced MSCs (iMSCs) generated by differentiating the induced pluripotent stem cells exerted anti-inflammatory and neuroprotective effects in a retinal ischemia-reperfusion model [87]. The protective effects were linked to immunomodulatory effects of iMSC as they reprogram mouse CD4-T cells into Foxp3 regulatory Tregs in vitro and to decrease retinal inflammation in vivo [87]. While MSCs are a relatively new way to combat retinal ischemia, MSC therapeutics are limited by low uptake rate and compatibility with the vitreous domain of the eye. Thus, it is crucial to consider molecular priming and paracrine extracellular vesicles to promote the efficiency of this therapeutic method, as discussed in the following section.

### 3.3. Preclinical Studies of MSCs-Exosomes in Ischemic Retina Models

Priming ASC with a combination of TNFα and INF-γ cytokine cocktail then injecting its CM in diabetic mice reduced vascular leakage and partially alleviated GFAP levels compared to controls [88]. Likewise, CM from MSC that were conditioned in a hypoxia-conditioned environment enhanced retinal function recovery compared to normoxic-conditioned medium [89]. When delivered 24 h after the ischemia injury, the hypoxic-preconditioned medium improved the retinal function, directly correlating these studies to clinically important therapies. Furthermore, a recent study using intravitreal delivery of CM from BM-MSCs reduced retinal pathological neovascularization in oxygen-induced retinopathy. This effect was attributed to Sema3E and reducing pathological levels of pro-inflammatory IL-17A [90]. CM is a rich source of extracellular vesicles that encompass a variety of molecular agents, nucleic acids, and proteins. In vitro, R28 retinal cells under stress from oxygen-glucose deprivation (OGD) exposed to MSC-derived EVs significantly minimized the rate of apoptosis and rescued cells from OGD stress [91]. In support of this, endocytosis of EVs, when delivered into the vitreous cavity, significantly reduced the levels of caspase-3 proteins in the retinal ischemia model [91]. Since a variety of miRNAs are found in EVs isolated from MSCs, few studies have explored the association of miRNA in exosomes in retinal ischemia. For example, delivering miRNA-17-3p via injected exosomes derived from human umbilical cord MSCs to diabetic mice resulted in reduced apoptosis and inflammatory molecules, including TNF-α, IL-6, and VEGF via targeting STAT1 [92]. In a separate study, MSC-derived EVs with miRNA-192 were explored to delay inflammatory response via targeting ITGA1, an integrin subunit receptor protein that is associated with the progression of Type 2 diabetes [93]. Taken together, transplantation of modified MSCs could be a novel strategy for the treatment of neurovascular degeneration in ischemic diseases. Exosomes isolated from human-MSC that are pre-conditioned in a hypoxic environment and delivered intravitreally into oxygen-induced retinopathy (OIR) mice resulted in lower, but not complete, restoration of retinal ischemia [94]. It is noteworthy that exosomes from human MSCs were well tolerated without immune suppression, and no immunogenicity was observed, paving the way for future allogeneic therapies. Recently, Mead et al. supported the use of small EVs derived from BM-MSC to combat RGC loss and visual deficits that arise from glaucoma [95]. In the study, mice were subject to pigmentary glaucoma, and at 6 months of age, reduced degradation of axons in the optic nerve was detected following EV injections [95]. In parallel, CM from ASC delivered intravitreally protected against activated microglia and Muller cell gliosis in a mouse model blast-related neurotrauma [96,97,98]. The MSCs-derived exosomes and their immunomodulatory potential were also reported in an experimental rat model of autoimmune uveitis through inhibiting migration of inflammatory cells to the retina [99].

### 3.4. Clinical Studies of MSCs in Ischemic Retinopathy

To date, there are no clinical studies evaluating the effect of MSCs in ischemic retina diseases; however, animal models support mechanisms for neuroprotection and vascular repair. MSCs participate in retinal repair in ischemic retinal injury models, homing into damaged areas and differentiating into endothelial cells, microglia, and astrocytes [100,101]. Patients underwent autologous Bone Marrow-Derived Stem Cell therapy within the Stem Cell Ophthalmology Treatment Study (SCOTS), an approved clinical study for the treatment of optic nerve and retinal diseases [102,103]. SCOTS and SCOTS II are large-scale clinical trials that include various optic conditions, including ischemic optic neuropathy, macular degeneration, and retinitis pigmentosa. Different clinical trials registered on clinicaltrials.gov are testing the efficacy and safety of MSC in retinal diseases (see Table 2). Shi et al. used BM-MSC in phase II randomized trial for patients with neuromyelitis optica (NCT02249676). The treatment includes bone marrow derived stem cells (BMSC) as retrobulbar, subtenon, intravitreal, intraocular, subretinal, or intravenous injections. Several preliminary results were published, favoring the BMSC treatment for the different ocular conditions with no complications or adverse effects [104,105,106,107]. 

**Table 2 ijms-23-00249-t002:** Clinical Trials that utilized MSCs in ischemic Retinopathy.

Study	Clinical Trials Identifier	Intervention	Delivery Method	Disease	Phase	Status	Reference
Stem Cell Ophthalmology Treatment Study (SCOTS)	NCT01920867	Autologous BM-MSC	Retrobulbar, subtenon, Intravenous, intravitreal, intraocular injections	Degenerative, ischemic or physical damage	N/A	Completed	[106]
Stem Cell Ophthalmology Treatment Study II (SCOTS2)	NCT03011541	Autologous BM-MSCs	Retrobulbar, subtenon, Intravenous, intravitreal, intraocular injections	Degenerative, ischemic or physical damage	N/A	Recruiting	[103]
Randomized trial for patients with neuromyelitis optica	NCT02249676	BM-MSC	IV infusion	neuromyelitis optica	phase II	Completed	

## 4. Wound Healing

### 4.1. Clinical Scope and Need for MSC Therapeutics

Wound healing is a complex process that involves many cell types, cytokines, and chemokines to restore the skin’s functionality. Chronic wounds impose a physical and psychological burden on patients and a cost burden. The economic burden is mainly due to amputations, ranging between 12,850-16,270 USD, excluding the cost of vascular and arterial leg ulcers [108]. Major contributors to the development and sustaining of chronic wounds are the failure in the cascade of the healing process that includes four phases: hemostasis, inflammation, proliferation, and remodeling (reviewed in [44]). In response to injury, platelet aggregation is activated, followed by the release of growth factors and chemotactic factors. Monocytes are then differentiated into macrophages during the inflammation phase, phagocytoses the tissue debris, bacteria, and the remaining neutrophils [109]. M1 macrophages have pro-inflammatory properties, expressing inflammatory cytokines and chemokines, including IL-1β, IL-12, and TNF-α. M1 macrophages are later transformed into M2 macrophages under the influence of T-helper2 (Th2) cytokines [110]. M2 subset secretes anti-inflammatory factors such as IL-10, TGF-β, and M2 markers (IL-1RA, CD163, and C-C motif chemokine 22) [110]. The transformation/polarization from M1 to M2 macrophages helps with the transition from the inflammation to the proliferation phase; failure to such a transition results in chronic wounds observed with diabetic wounds [111]. Many treatments for delayed wound healing have been developed, including various wound protective dressings such as basic wound contact dressings, advanced wound dressings, antimicrobial dressings, and specialist dressings (reviewed in [112]). Studies showed superior wound healing with hydrogel and foam dressings compared with basic wound contact materials and higher healing with hydrocolloid-matrix dressings compared with other dressing types [112]. Other modalities promoting wound healing are systemic antibiotics usage and hyperbaric oxygen therapy (reviewed in [113]. Nevertheless, a significant portion of these wounds remains refractory to the standard treatment. Accumulating evidence shows that MSCs are a promising candidate for treating chronic non-healing injuries. They exert their therapeutic effect either by differentiation into other cell types or by secreting various factors (reviewed in [114]). The following section will discuss examples of modulating MSCs and utilization of cells and or exosomes in wound healing models.

### 4.2. Preclinical Studies of Modulated MSCs in Wound Healing Models

Since macrophages play a pivotal role in both pro-inflammatory and anti-inflammatory phases during the healing process, the effect of MSC injections on macrophage polarization has been extensively studied. In vitro, co-culturing macrophages with human gingival MSC facilitated acquiring an anti-inflammatory M2 evident by increased expression of CD206 and the anti-inflammatory IL-10, IL-6, and decreasing pro-inflammatory TNF-α expression [115]. In vivo, systemically infused MSCs homed to the excisional full-thickness wound site, interacted with host macrophages, promoted them toward M2 polarization, and significantly enhanced angiogenesis, deposition of extracellular matrix (ECM), and wound repair [115]. Fibroblasts appear during wound healing as they migrate from neighboring cells to the wound bed to release cytokines and synthesize the ECM resulting in decreased wound size [116]. These effects were attributed to enhanced expression of VEGF, basic FGF (bFGF), and TGF-β that induced fibroblasts proliferation and collagen synthesis [116]. Murine burn model injected with human umbilical cord (hUC)-MSCs demonstrated higher neovascularization and VEGF, collagen I, and III than control groups [117]. Diabetic wounds are characterized by diminished cell recruitment, impaired collagen matrix formation, and lack of growth factors. Shi et al. showed that systemically injected hUC-MSC accelerated the wound healing rate of diabetic foot ulcers in mice. These effects could be attributed in part to enhanced levels of VEGF, bFGF, and HGF, as well as epithelialization, granulation tissue formation, and collagen deposition [118]. Pretreatment of MSCs with salidroside significantly enhanced the effect of MSCs in promoting wound closure in diabetic mice [119]. These effects were attributed to restoring the expression of HO-1, FGF2, and HGF and suppression of intracellular ROS levels in MSCs, thereby lowering the apoptosis rate and enhancing MSC’s survival rate [119].

Since inflammation of the wound bed is the leading cause of hypertrophic scar formation, attempts to balance pro- and anti-inflammatory of MSCs were pursued. Umbilical cord-MSCs (UC-MSCs) hold immunomodulatory effects on the immune cells and inflammatory process during wound healing. UC-MSCs migrated into the wound and remarkably decreased the infiltrated inflammatory cells and IL-1, IL-6, TNF-α and increased levels of anti-inflammatory IL-10 in severe burn wound model [117]. Similarly, MSCs transplantation could suppress secondary inflammatory reactions after burn injury [120]. Overexpression of IL-10 in BM-MSCs resulted in small scar area and height in a rabbit ear hypertrophic scar model [121]. These effects were associated with decreasing expression of TNF-α, IL-6, IL-1β, as well as collagen I and α-SMA in the group receiving IL-10-modified BM-MSCs [121]. 

In contrast, Munir et al. have demonstrated that priming MSCs with lipopolysaccharide (LPS) to mimic infected wound environment has beneficial effects on neutrophil activation in murine wound model [122]. The results showed the upregulation of IL-1β and IL-8 chemokine that recruits and activates neutrophils and accelerates wound healing. The study has also shown that Toll-like receptor (TLR) was crucial for the neutrophil activation as the LPS primed, TLR-silenced MSC failed to activate neutrophils and subsequently didn’t accelerate wound healing compared to the controls [122]. Chen et al. investigated the effect of hypoxia-preconditioning of human BM-MSCs in vitro and in vivo [13]. Hypoxia upregulated mRNA expression of bFGF, IGF-1, VEGF-A, TGF-β when compared to non-hypoxic MSCs. The wounds treated with hypoxia-preconditioned BM-MSCs showed enhanced blood vessel formation that coincided with increased expression of anti-fibrotic collagen-III and decreased expression of the profibrotic collagen-I [13]. Delivery of murine BM-MSC by microsphere accelerated the wound closure and enhanced the epithelial and collagen organization. Later, the profibrotic drive was decreased, evidenced by the downregulation of collagen I and α-SMA [123]. 

### 4.3. Preclinical Studies of MSCs-Exosomes in Wound Healing Models

Preconditioning MSCs with various stressors has been reported to enhance MSC-derived exosomes’ biogenesis and therapeutic potential. A recent study showed superior effects of preconditioning with thrombin on EV production when compared to other regimens, including LPS, H_2_O_2_, or hypoxia [124]. Thrombin preconditioning enhanced angiogenin, angiopoietin-1, HGF, and VEGF levels and promoted endothelial proliferation and alignment of capillary-like tube structures compared with naïve MSCs EVs controls [124]. Conversely, Ti et al. demonstrated that exosomes from LPS-pretreated human UC-MSC promoted M2 macrophage activation when cocultured with human monocytic cell line THP-1. Preconditioning with LPS resulted in higher levels of IL-10 and TGF-β and increased density of M2 macrophages, and reduced density of M1 macrophages [125]. In vivo, injection of these LPS-pretreated exosomes promoted angiogenesis, increased wound healing rates, and increased M2 macrophages expansion but decreased M1 macrophages and inflammatory cells infiltration in streptozotocin-induced diabetic rats [125]. Overexpression of the miR-375 in BM-MSC from mice induced expression of HGF, an anti-scar factor and reduced levels of tissue inhibitor of metalloproteinases-1, α-SMA, TGFβ, collagen I, and Fibronectin indicating the suppression of fibroblasts transition to myofibroblasts [126]. 

Long noncoding RNAs (lncRNAs), an exosomal component, are involved in the transcriptional regulation of several pathways, including the inflammatory response via mitogen-activated protein kinase. Several studies have documented the beneficial role of the (lncRNA) H-19 in wound healing in diabetic foot ulcers [92,127,128]. Exosomes carrying lncRNA H19 cocultured with fibroblasts resected from patients with diabetic foot ulcer promoted fibroblasts proliferation and migration and suppressed apoptosis. In mice with diabetic foot ulcers, injecting the MSC-exosomes containing the H19-overexpressed into the surrounding tissues of the wound has resulted in increased IL-10 levels and lowered the IL-1β and TNF-α levels [129]. Moreover, angiogenesis has been improved by increasing the expression of VEGF, TGF-β1, α-SMA, and collagen-I. Another type of non-coding RNA, circular RNA (circRNA), has been proven to play a role in regulating the pathogenesis, progression, and complications of diabetes in vitro and in humans and mice models reviewed in [130,131]. 

### 4.4. Clinical Studies of MSCs in Wound Healing

Several clinical studies and case reports addressing the role of mesenchymal cells in wound healing have been published, and more trials are still in the recruitment phase (see Table 3). A pilot study in 2009 included patients with Beurger disease and diabetic foot ulcers, and the cases received autologous BM-MSCs. The treatment proved to be effective as the ulcer size decreased and the pain and the pain-free walking distance increased. The biochemical parameters such as liver and kidney functions showed no significant change before and after MSCs implantation, indicating the safety of the BM-MSCs [132]. Another group tested the impact of hUC-MSC on the levels of inflammatory markers in diabetic patients with a foot injury. Results showed that TNF-α, IL-6, and CRP were lower than pre-treatment levels. They also pointed that MSCs therapy can have beneficial effects on pancreatic β cells [133]. In parallel, Nojehdehi et al. reported the immune-modulatory potential of MSCs-derived exosomes in a mouse model of type-1 DM [134]. Another open-label clinical trial has tested the healing effect on non-healing ulcers after hUC-MSCs transplantation in diabetic patients with foot injury [135]. Results showed improvements in skin temperature, ankle-brachial pressure index, and transcutaneous oxygen tension as early as 3 months that led to gradual or complete ulcer healing compared to controls.

**Table 3 ijms-23-00249-t003:** Clinical Trials that utilized MSCs in wound healing.

Study	Clinical Trials Identifier	Intervention	Delivery Method	Disease	Phase	Status	Reference
Stem Cell Therapy to Improve Burn Wound Healing	NCT02104713	Allogeneic MSCs	Topical Application	Skin burns 2nd degree	Phase I	Completed	
Safety and Exploratory Efficacy Study of Collagen Membrane with Mesenchymal Stem Cells in the Treatment of Skin Defects (SEESCMMSCTSD)	NCT02672280	MSC with Medical Collagen Membrane	Topical application	Wounds Diabetic Foot Ulcers Burns	Phase I Phase II	Unknown	
Human Placental Mesenchymal Stem Cells Treatment on Diabetic Foot Ulcer	NCT04464213	Human placental MSC gel	Topical application	Diabetic Foot Ulcer	Phase I Phase II	Recruiting	
Phase I, open-label safety study of umbilical cord lining mesenchymal stem cells (corlicyte^®^) to heal chronic diabetic foot ulcers	NCT04104451	Expanded UC-MSC (Corlicyte^®^)		Chronic Diabetic Foot Ulcers	Phase I	Recruiting	
Clinical Application of MSC Seeded in Chitosan Scaffold for Diabetic Foot Ulcers	NCT03259217	ASC seeded in Curcumin loaded into collagen-alginate	Topical application	Diabetic Foot Ulcers	Phase I	Unknown	
Comparison of Autologous MSC and Mononuclear Cells on Diabetic Critical Limb Ischemia and Foot Ulcer	NCT00955669	MSCs or MNCs transplantation	Intramuscular injection	Diabetic Foot	Phase I	Completed	[136]
Allogeneic ABCB5-positive Stem Cells for Treatment of DFU “Malum Perforans	NCT03267784	allogeneic ABCB5-positive MSCs	Topical application	Diabetic Neuropathic Ulcer	Phase I Phase II	Completed	[137]
Therapeutic Potential of Stem Cell Conditioned Medium on Chronic Ulcer Wounds	NCT04134676	hWJ-MSC conditioned media	Topical application	chronic skin ulcers	Phase I	Completed	
Safety of MSC Extracellular Vesicles (BM-MSC-EVs) for the Treatment of Burn Wounds	NCT05078385	AGLE-102 (BM-MSCs- derived EVs)	Direct application	2nd degree burn	Phase I	Not yet recruiting	

Several techniques for MSC delivery have been proposed for efficient delivery and maximum effect. A case report of a non-healing diabetic ulcer patient was treated with a combination of autologous BM-MSCs along with autologous skin fibroblasts on a biodegradable collagen membrane (Coladerm). The combination was applied on days 0, 7, and, 17 and the combination showed increased wound healing and vascularization after 29 days [138]. La fosse et al. tested the effectiveness of ASC seeded onto a human decellularized collagenic scaffold on three patients. They reported favorable adhesion and spreading compared to injections. It has limited tendencies to develop tumors and no adverse effects up to 22 months. Moreover, they eliminate the need for painful, repetitive injections in the fibrotic region [139]. Falanga et al. documented delivering autologous BM-MSCs topically using fibrin spray. The cells remained viable and could migrate to the wound after application, showing promising results in both acute and chronic wounds [140]. A case report of a thermal burn patient, mostly full-thickness, showed increased wound healing with no hypertrophic scars after autologous administration of amniotic and hUC-MSCs [141]. Another burn case was treated by skin grafting and fibroblast-like from hBM-MSCs, resulting in pain relief, decreased hypersensitivity, and active epithelialization [142]. 

## 5. Peripheral/Critical Limb Ischemia (PLI/CLI)

### 5.1. Clinical Scope and Need for Therapeutics

Peripheral/critical limb ischemia (PLI/CLI) is the most severe form of peripheral arterial disease (PAD) that is characterized by ischemia in the lower extremities due to narrowing of arteries with atherosclerotic plaque accumulation [141]. PAD is increasingly recognized as an important cause of cardiovascular morbidity and mortality that affects >230 million people worldwide [143]. It is estimated that 25% of patients diagnosed with PLI will die within 1 year, and an additional 30% will receive limb amputation [144]. The symptoms usually experienced by patients include pain at rest, non-healing ulcers, and tissue necrosis with gangrene [145]. Thus far, there is no pharmacotherapeutic agent available for the treatment or prevention of PAD [144]. Treatment options other than limb amputation are restricted to surgical revascularization, which is not recommended in patients with severe co-morbidity, sepsis, limb gangrene, or non-ambulatory individuals [146]. Consequently, strategies implementing cell-based therapies have emerged to combat ischemia in patients with no other treatment options in terms of tissue damage repair and regeneration [147]. In 2008, Crisan et al. first established the perivascular origin for MSCs and their role in blood vessel stabilization [148]. In addition, MSCs are well known to be home to sites of ischemia and secrete a broad spectrum of pro-angiogenic and immunomodulatory factors to support angiogenic and arteriogenic processes [149]. Thus, MSCs can stabilize blood vessels and support their maturation. 

### 5.2. Preclinical Studies of Modulated MSCs in Critical Limb Ischemic Models

BM-MSCs improved limb function and appearance, reduced the incidence of auto-amputation, and attenuated muscle atrophy and fibrosis in a mouse model of distal femoral artery ligation. Analysis of MSCs-CM revealed the release of VEGF, bFGF, PlGF, and MCP-1 [150]. Adipose-derived MSCs showed better recovery of blood flow in a mouse model of CLI as compared to BM-derived-MSCs, an effect that was attributed to upregulated expression of MMP3 and MMP9 [151], whereas Nammian P et al., 2021, reported better recovery and more efficient angiogenesis in the CLI mouse model upon allogeneic transplantation of BM-derived MSCs as compared to adipose tissue-derived MSCs [152]. The potential of BM-derived MSCs was further supported by Yao Z et al., findings where IM injection of BM-MSCs-derived endothelial cells into GC muscles of ischemic mouse limbs restored blood flow and accelerated angiogenesis [153]. MSCs differentiated from ESCs [154] or iPSCs [101] combated hindlimb ischemia as well, with superior activity as compared to BM-MSCs [101]. Specific cell subsets of human MSCs that express VCAM-1 have been shown to exert stronger therapeutic potential in the ischemic hindlimb of nude mice [155]. A recent study reported that hUC-MSCs or BM-MSC relieved hind limb ischemia in mice, possibly through activation of ERK and PI3K-Akt pathways thus, promoting angiogenesis [156]. Gao W et al., 2019, demonstrated the effectiveness of UC-MSCs in patients with CLI through anti-inflammatory, immunomodulation, and enhanced wound healing [157]. MSCs regulated macrophage differentiation toward the M2 phenotype, thus alleviating inflammation and attenuating hindlimb ischemia [158]. 

Strategies have been developed to enhance the angiogenic capacity of MSCs for PLI because single-cell treatments alone might not be sufficient to treat the severe disease effectively. Examples include inhibition of aldehyde dehydrogenase and retinoid signaling [159]. Netrin-1-treated MSCs promoted MSCs revascularization potential in a rat model of peripheral limb ischemia by increasing the level of VEGF [160]. Another study showed that JH4, an inhibitor of lamin A-progerin binding, improved the survival of human adipose tissue-derived MSCs in oxidative stress conditions, improving blood reperfusion recovery and limb salvage [161]. Promotion of angiogenic capacity of MSCs through treatment with melatonin was also reported [162]. Moreover, concurrent delivery of the antioxidant; N-acetylcysteine with aggregates of HUVECs and cord blood-derived MSCs enhanced cell adhesion, retention, and survival in a mouse model of hindlimb ischemia [163]. Treatment of human AD-MSCs with TGF-1 was shown to increase the expression of vascular SMC-like ion channels and the differentiation of hASCs into contractile vascular SMCs [164]. Conversely, ischemic preconditioning was shown to enhance angiogenic potential and decrease the inflammatory immune response of human umbilical cord derived MSCs in the ischemic hindlimb of immunodeficient mice [165]. In addition, genetic modification of human placenta-derived MSCs with FGF-2 and PDGF-BB enhanced neovascularization in a unilateral model of hindlimb ischemia induced in New Zealand White rabbits [166]. 

Modern techniques have been developed for MSCs engineering in the hope of enhancing their vascular therapeutic potential. Ishii M et al. created an MSC sheet using magnetic nanoparticle-based tissue engineering technology [167]. Zhao N et al. demonstrated another method through co-transplantation of MSCs with a synthetic hydrogel formed by conjugating C domain peptide of insulin-like growth factor-1 onto chitosan, resulting in an enhanced therapeutic potential for hindlimb ischemia [168]. Meanwhile, Lee JH et al. utilized the spheroid culture of adipose-derived MSCs to preserve their microenvironment in a murine hindlimb ischemia model. MSCs spheroids were generated by suspension culture for 3 days, with a consequent increase in their sizes in a time-dependent manner. MSCs spheroids promoted MSCs bioactivities via secretion of angiogenic cytokines, preservation of ECM components, and regulation of apoptotic signals [169]. 

### 5.3. Preclinical Studies of Paracrine Factors of MSCs in Critical Limb Ischemic Models

As discussed previously, there is increasing evidence that the therapeutic effect of stem cells is regulated primarily through a paracrine mechanism rather than direct engraftment and trans-differentiation. In the context of PLI, the therapeutic potential of the paracrine secretome of MSCs has been investigated. CM of MSCs promoted the angiogenic activity of cord blood derived endothelial colony-forming cells (ECFCs) and their capacity to recover blood perfusion in hindlimb ischemia through up-regulation of sphingosine kinase 1 (SphK1) expression and activity and sphingosine-1-phosphate receptor 1 (S1P1) dependent pathway [170]. Exosomes secreted by human-induced pluripotent stem cell-derived MSCs attenuated limb ischemia in mice and promoted angiogenesis [171]. Anderson JD et al., 2016, performed a proteomic analysis that revealed a robust profile of angiogenic paracrine effectors in MSCs-derived exosomes that enhanced their potential for treating ischemic diseases. These include platelet-derived growth factor, epidermal growth factor, fibroblast growth factor, and most notably nuclear factor-kappaB (NFkB) signaling pathway proteins [172]. More recently, Yan B et al., 2020, demonstrated that the therapeutic potential of MSCs-derived exosomes for hindlimb ischemia is regulated through the miR-421/FOXO3a pathway, preventing pyroptosis and repairing ischemic muscles [173]. Micro-vesicles can be an additional contributor to MSC paracrine-mediated vascular therapeutic potential. Micro-vesicles derived from human umbilical cord MSCs that were stimulated by hypoxia promoted angiogenesis in vitro and in vivo in a rat model of hindlimb ischemia [174]. ASCs-induced angiogenesis was attributed to micro-vesicle transport of miRNA-31 and suppression of HIF-1α [175]. 

Modern techniques have been developed to augment MSCs secretome. Decellularized bio-scaffold is an emerging modality that has generated great interest to enhance tissue regeneration in PLI [176]. In brief, the scaffold, being enriched with structural ECM components, can support cell attachment, infiltration and constructive tissue remodeling in vitro and in vivo [177,178]. Of note, decellularized adipose tissue (DAT) scaffolds administered SC into Wistar rat model were well-tolerated and integrated into the host tissues, supporting angiogenesis and adipogenesis. DAT-scaffolds gradually resolved over the course of 12 weeks, demonstrating their potential for wound healing and soft tissue regeneration [179]. Conversely, human MSCs that were contained in biocompatible alginate microcapsules demonstrated augmented paracrine proangiogenic activity with no incorporation of actual cells into the host tissue of a murine model of hindlimb ischemia [104]. Another method for enhancing the stability and retention of MSCs-derived exosomes was developed by Zhang K et al., 2018, through exosomes incorporation with chitosan hydrogel that resulted in augmented therapeutic effect for hindlimb ischemia [180]. 

### 5.4. Clinical Studies of MSCs in PLI

Several clinical studies have investigated the potential of MSCs in patients suffering PLI (see Table 4). A case report published in 2010 showed that autologous IV infusion of human BM-MSCs fostered revascularization and reduced skin necrosis in a female patient with systemic sclerosis who developed acute gangrene of the upper and lower limbs [181]. RESTORE-CLI trial reported by Powell RJ et al., 2012, demonstrated the therapeutic potential of cultured BM-MSCs and hematopoietic progenitor cells for PLI patients with no options for revascularization. Improved mortality rates and gangrene were observed with no differences in amputation rate [182]. Another study conducted by Lu D et al., 2011, demonstrated the superiority of BM-MSCs to BM-MNCs for treating diabetic CLI patients and foot ulcers in terms of increasing lower limb perfusion and promoting foot ulcers, with no differences observed in amputation rates [183]. Gupta PK et al., 2013, demonstrated the safety and efficacy of IM-injected BM-MSCs at a dose of 2 million cells/kg body weight in patients with established PLI. Improved parameters included rest pain scores, ankle-brachial pressure index, and ankle pressure but not amputation rates [184]. The feasibility and safety of autologous ASCs transplantation in patients with PLI not suitable for revascularization were further established in a phase-I trial developed by Bura A et al., 2014 [185]. Improved ulcer evolution and wound healing with no safety problems were also reported [185]. 

It is well known that the therapeutic potential of MSCs is affected by the disease state. For example, MSCs derived from patients with atherosclerosis adopt a pro-inflammatory secretome via increased secretion of IL-6, IL-8, and MCP-1, reversing the normally immunosuppressive nature of MSCs [186]. This fact explains compromised function in clinical trials that utilize autologous cell therapy [149]. However, Gremmels H et al., 2014, reported equivalent neovascularization capacity of MSCs derived from CLI patients and healthy controls in a murine hindlimb ischemia model, supporting autologous stem cell therapy for CLI patients [187]. Agreeing with this finding, Brewster L et al., 2017, also demonstrated that MSCs expanded from CLI patients demonstrate the desired angiogenic activity compared with their healthy counterparts. In addition, MSCs from CLI patients can be sufficiently expanded with the novel culture media supplement, pooled human platelet lysate, to improve successful delivery of autologous MSCs to patients with CLI [188]. These controversial results may be due to the different MSCs sources, disease statuses, and/or angiogenic markers. Further investigation will be required to evaluate the angiogenic capacity and potential clinical applications of human MSCs derived from patients with ischemic vascular diseases.

A meta-analysis of randomized placebo-controlled trials for the use of BM-derived cell therapy in CLI patients demonstrated that cell therapy significantly improved ankle-brachial index (ABI), resting pain, and pain-free walking time, with no advantage of stem cell therapy on the primary outcome measures of amputation, survival, and amputation free survival. The analysis involved 10 randomized, placebo-controlled trials with a total of 499 CLI patients [189]. Another more recent meta-analysis that included 19 randomized controlled trials with 837 CLI patients treated with autologous stem cell therapy concluded a modest reduction in the risk of amputation by 37%, improved amputation free survival by 18%, and improved wound healing by 59% with no effect on mortality rates [190].

## 6. Limitation of MSC Therapeutics and Future Perspectives

MSCs hold great potential for therapy, and as depicted in Figure 1, the regenerative benefit of the MSCs could be contributed partially to their ability to be differentiated to other cell types but largely mediated via paracrine effects. In addition, MSC-derived secretome and exosomes and the release of trophic factors and cytokines exert neuroprotective, angiogenic, anti-inflammatory, and immune-modulatory roles. As discussed in preclinical and clinical studies, the potential of MSCs and their-derived exosomes is promising and quite effective for diverse ischemic vascular disease states such as ischemic heart disease, ischemic retina, critical limb ischemia, and wound healing. Additional disease states that are beyond the current review include acute respiratory distress syndrome, coronavirus pneumonia, acute ischemic stroke, Alzheimer’s Disease [4]. 

Some of the evolving problems associated with MSC-derived therapeutics include manufacturing challenges such as purity and consistency. Further, low cell yields and increased costs related to manufacturing coupled with poor engraftment in vivo have driven the cell-based therapies towards cell-free treatments. A dominant part of the literature currently describes that MSCs primarily elicit their benefits through EVs [191,192,193]. However, as the research evolving, several limitations of such EV-based MSC therapeutics have emerged to effectively translate such studies. EVs are exceedingly more complex than the MSCs they are released from, as they participate in triggering intracellular cell signaling by releasing a variety of proteins and miRNAs [194]. Future studies are necessary to de-code this complex nature of EVs to better understand their mechanism of action on the target tissue. To avoid the confounds of manufacturing limitation of MSCs, efforts to bioengineer MSCs to ensure expansion and production of homogenous MSCs in larger amounts is ongoing and represents an active area of research. Either immortalized MSCs using the human telomerase reverse transcriptase (hTERT) method, induced pluripotent stem cell (iPSC) strategy or a clustered regularly interspaced short palindromic repeats (CRISPR)/Cas9-based strategy to derive MSC-EVs likely serve as a reliable cell source to develop consistent products [194,195]. Concerns do exist about iPSCs regarding tumorigenesis, but pro-tumor formation from MSCs via iPSCs have been shown to be insignificant in previous studies [196]. Another field of growing interest related to improving MSC therapeutics is exosome engineering. By loading the desired protein(s) and/or miRNA, exosome engineering methods to artificially enhance EVs therapeutic benefit to combat specific diseases is another active area of research [194]. Although the future of EV based MSC therapeutics seems bright, there are several additional challenges including accurate quantification of EVs, their specific mechanism of action and GMP compliance in manufacturing and product consistency needs further fine tuning [197].

**Table 4 ijms-23-00249-t004:** Clinical Trials that utilized MSCs in peripheral limb ischemia.

Study	Clinical Trial Identifier	Intervention	Delivery Method	Disease	Phase	Status	Reference
Cellular therapy with Ixmyelocel-T to treat critical limb ischemia: RESTORE-CLI trial		BM-MSC	Intramuscular injections	PLI patients with no options for revascularization	Phase II	Completed	[182]
Comparison of BM-MSC for treatment of diabetic critical limb ischemia and foot ulcer:		IM BM-MSCs IM BM-MNCs	Intramuscular injection	Diabetic patients with bilateral CLI and foot ulcer			[183]
The safety and efficacy of allogeneic BM-MSC in critical limb ischemia	NCT00883870	BM-MSCs	Intramuscular injection	Critical limb Ischemia	Phase I Phase II	Completed	[184]
The use of autologous cultured ASC- to treat patients with non-revascularizable CLI	NCT01211028	Adipose-derived stroma/stem cells	Intramuscular injection	Non-revascularizable critical limb ischemia	Phase I	Completed	[185]
BM-derived Cell Therapy in Critical Limb Ischemia		Bone marrow-derived stem cells	-	Critical limb ischemia	Meta-analysis		[189]
Autologous Cell Therapy for Peripheral Arterial Disease:		Randomized, Nonrandomized, and Noncontrolled Studies	-	Critical limb ischemia Intractable peripheral arterial disease	Systematic Review and Meta-Analysis		[190]
Safety and Efficacy of Allogeneic Adipose Tissue-MSCs in Diabetic Patients with CLI	NCT04466007	Low and high dose IM Allogeneic ASC	Intramuscular injection	Limb Ischemia Diabetic Foot	Phase II	Recruiting	

## Figures and Tables

**Figure 1 ijms-23-00249-f001:**
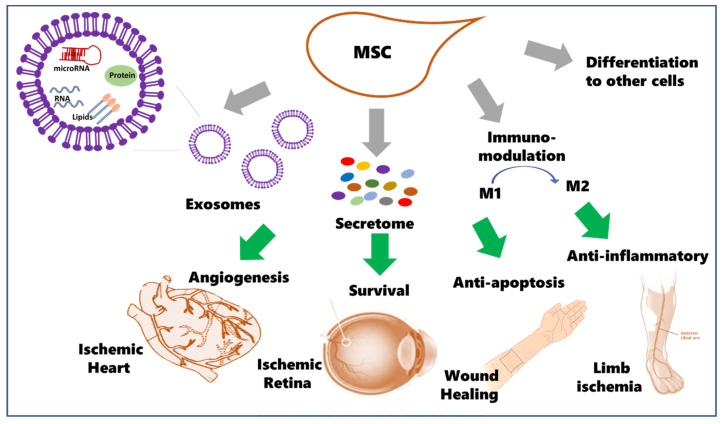
Postulated mechanisms for therapeutic effects of mesenchymal stem cells (MSCs) including differentiation to other cell types, immunomodulation from M1 to M2 as well as the main paracrine effect of MSCs-derived secretome (growth factors, cytokines and exosomes). Exosomes from MSCs contain multiple proteins, lipids, RNAs (mRNA, miRNA, ncRNA). MSCs secretome obtained for therapeutic application can be tailored or modified to the desired cell-specific effects. Therapeutic effects of MSCs-were examined in ischemic heart, retina, wound healing and critical limb ischemia disease states.

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
