# Peer review of "Modulation of Mesenchymal Stem Cells for Enhanced Therapeutic Utility in Ischemic Vascular Diseases"

_ijms, 2021, doi:10.3390/ijms23010249_

Round 1
Reviewer 1 Report
This is a well written review aiming to present the beneficial potential of modulating the content of vesicles secreted by mesenchymal cells. Modification of the secretome of these vesicles is a great challenge nowadays. I would recommend slight modifications in the text to emphasize the relevance of several studies: see for example reference 75. The relevance of modulation of cells that secrete the vesicles appears very "shyly" in the text. Please, modify the legend of the figure to clearly introduce the idea of secretome modulation.
Author Response
-This is a well written review aiming to present the beneficial potential of modulating the content of vesicles secreted by mesenchymal cells. Thank you!
- Modification of the secretome of these vesicles is a great challenge nowadays. I would recommend slight modifications in the text to emphasize the relevance of several studies: for example, reference-75. The relevance of modulation of cells that secrete the vesicles appears very "shyly" in the text. Modified as suggested- note that this reference is now 78.
- Please, modify the legend of the figure to clearly introduce the idea of secretome modulation. Modified as suggested.
Reviewer 2 Report
In the present review of “Modulation of mesenchymal stem cells for enhanced therapeutic utility in ischemic vascular diseases”, Elshaer and colleagues summarized the use of MSCs and other factors derived from the cells for treating various ischemic vascular diseases. The paper is well-written and interesting. My comments are stated below:
- It will be better for the readers to add some more details about each diseases and to briefly summarize the current treatment for each diseases. Also description about the advantages of MSC-based therapy compared to other treatment might be helpful as well.
- The authors should also discuss the limitations of MSC or MSC-derived secretome therapy and the potential future directions of using them to treat the diseases.
- In the clinical trial table, please add the delivery methods for all cases.
Author Response
In the present review, Elshaer and colleagues summarized the use of MSCs and other factors derived from the cells for treating various ischemic vascular diseases. The paper is well-written and interesting. Thank you!
My comments are stated below:
- It will be better for the readers to add some more details about each disease and to briefly summarize the current treatment for each disease. Modified and new sections are added (highlighted).
Also, description about the advantages of MSC-based therapy compared to other treatment might be helpful as well. The advantages of MSC-therapy, are discussed under section “potential of stem cell therapy on page-1. In addition, a transition sentence about MSC is added to the available treatment.
- The authors should also discuss the limitations of MSC or MSC-derived secretome therapy and the potential future directions of using them to treat the diseases. This new section is added on page-16, 17.
- In the clinical trial table, please add the delivery methods for all cases. Modified as suggested.